# Effects of Digital Transformation on Environmental Governance of Mining Enterprises: Evidence from China

**DOI:** 10.3390/ijerph192416474

**Published:** 2022-12-08

**Authors:** Chaohui Xu, Xingtong Chen, Wei Dai

**Affiliations:** 1School of Economics and Management, Hubei University of Science and Technology, Xianning 437000, China; 2School of Business Administration, Southwestern University of Finance and Economics, Chengdu 610000, China; 3School of Economics and Management, Hubei Polytechnic University, Huangshi 435003, China

**Keywords:** digital transformation, environmental-protection investment, media supervision, accounting data comparability, mining enterprises

## Abstract

Digitization offers fresh impetus to the transformation and upgrading of mining enterprises, while on the other hand, the rapid development and broad application of digital technologies make the environmental governance of mining enterprises the most important themes of theoretical research and practical exploration. In this paper, A-share companies listed between 2007 and 2020 are taken as samples to study the influence of digital transformation on the environmental governance of mining enterprises, and its relative acting paths. Our main research methods are multiple linear regression analysis, the panel fixed-effect model and the intermediary effect model. The results show that digital transformation significantly improves the environmental governance of mining enterprises, which is still tenable even after going through a series of endogeneity and robustness tests. It is found via the path test that, by strengthening the supervision of the media, the digital transformation performed in mining enterprises helps improve their environmental governance level, but the comparability of the accounting data shows no significant mediation effect between digital transformation and environmental governance. The heterogeneity test found that the promotion of digital transformation in environmental governance was significant only in non-state-owned enterprises, large-scale enterprises, and mature-growth enterprises. The findings enrich studies on the economic consequences and the environmental governance influences brought by mining enterprise’s transformation based on advanced technologies. This provides an important reference and is of great heuristic significance in promoting digital transformation and strengthening the environmental governance of mining enterprises.

## 1. Introduction

The mining enterprise has unique features that set it apart from other enterprises. On the one hand, the mining industry is fundamental for the national economy and a key source of energy, industrial raw materials, and agricultural production means, playing an important role in securing state resources, promoting national economic growth, and facilitating regional economic construction. On the other hand, mining enterprises strongly rely on natural resources and could pollute or cause environmental damage to the air, water, and soil. Since China’s economic reforms were implemented in 1976, the country has witnessed a rapid growth of the transport network along with the rise in GDP (Magazzino and Mele, 2020) [1]. In fact, the industry of mining has a long history in China, and its extensive development has led to long-term solutions for a series of social problems (Udemba et al., 2020) [2], such as the severe pollution of three types of waste (waste gas, water, and residuals) and the inadequacy of resource reserves, etc. The social problems caused by this extensive development are catapulting China into an era of high pollution that is causing severe environmental damage. In particular, the incidence of mining-induced environmental pollution, ecological damage, and pollution in specific areas has received widespread scrutiny from the public. According to the report of Guangming Net, the number of cases concerning the environment and resources heard by Chinese courts reached 253,000 in 2020, and in 2021, according to the Xinhua News Agency, this figure reached 265,300. In this context, China proposes the concept of building a beautiful China and constructing ecological civilization, indicating a new type of mine construction compatible with an ecological civilization. In fact, reducing pollutant emissions has a positive effect on sustainable economic development (Mele and Magazzino, 2020; Magazzino et al., 2020) [3,4]. Therefore, it is essential to study the environmental governance of mining enterprises and its significance.

According to the theory of organizational legitimacy, it is difficult for a social organization to survive and develop if it cannot meet the requirements of its implicit social contract. In order to cope with the public pressure generated by this implicit contract and recognize environmental legitimacy, enterprises must demonstrate more positive environmental management behavior (Cho and Patten, 2007) [5]. However, due to the separation of corporate control and ownership, agency problems arise, the interests of managers and shareholders are inconsistent, and managers are guided by the maximization of their own interests (Wang and Xu, 2018) [6]. Enterprises implement performance pay for managers: managers’ pay is linked to the company’s performance. Meanwhile, environmental investment increases the cost of enterprises, reduces the short-term performance of enterprises, and managers’ pay is reduced. As a result, managers lack the incentive to promote environmental governance. The information interaction system between enterprises and stakeholders and the information disclosure system of enterprises are not perfect, resulting in the information asymmetry between the two sides. In the case of information asymmetry, managers with opportunistic tendencies tend to reduce investment in environmental protection and increase the illegal discharge of pollutants. However, under the condition of digitalization, the high versatility and penetration of digital technology will lead to a zero distance between enterprises and stakeholders. In particular, when digitalization is deeply embedded in the operation management and business system of enterprises, the information disclosure system of enterprises to stakeholders will also be comprehensively reshaped in the digital context (Hinings et al., 2018) [7]. Digitalization can greatly reduce the information asymmetry and interaction cost between mining enterprises and stakeholders, reduce the self-interest tendency of managers driven by opportunism, restrict the discretion of mining enterprises in terms of pollutant discharge, and promote managers to face the environmental governance of mining enterprises. In the era of digital economy, the development and application of digital technology can play a positive corporate governance effect, which in turn affects the environmental governance of mining enterprises.

The purpose of this paper is to study the influence of digital transformation on environmental governance of mining enterprises and its internal mechanism. The innovative points of this paper can be summarized into three aspects. Firstly, the current literature mainly focuses on the impact of digital transformation on enterprise changes in management and performance; therefore, there is a lack of studies regarding the influence of enterprise governance on environmental protection. In the fusion development context of the digital economy and environmentally friendly development of mining enterprises, the relationship between digital transformation and environmental governance is explored, and digital transformation is combined with environmentally friendly development, in order to expand studies on the economic consequences of digital transformation. Secondly, most of the existing literature focuses on factors affecting environmental governance from the perspective of internal governance structure and management heterogeneity of the enterprise, rarely paying attention to the influences of data processing facilities of the enterprise on its environmental governance. Therefore, in this paper, the influence of digital transformation on environmental governance is studied from the perspective of the enterprise’s strategic digital transformation, which enriches research on the influence of the internal environment of an enterprise on environmental governance. Thirdly, the internal mechanism of the digital transformation of mining enterprises that affects environmental governance is discussed in this paper. Micro-tests were applied based on the large samples to unpack the theoretical black box regarding the non-economic performance of the mining enterprises, which are empowered by digital transformation. This provides a theoretical reference for encouraging mining enterprises to be more socially responsible in the digital era.

## 2. Literature Review

### 2.1. Economic Effects of Digital Transformation

Digital transformation can help enterprises improve production efficiency and performance but also has potential adverse effects (Yeow et al., 2017) [8], which lead to both success and failure in the digital transformation of enterprises (Lucas and Goh, 2013) [9].

The positive role of digital transformation is mainly reflected in production efficiency, performance, market position and dynamic capabilities. With regard to the impact of digital transformation on enterprise production efficiency, digital transformation has improved the total factor productivity of enterprises by improving their innovation ability, optimizing human capital structure, reducing costs, and promoting the integrated development of advanced manufacturing and modern service industries (Louridas and Ebert, 2016) [10]. With regard to the impact of digital transformation on enterprise performance, through digital product innovation, enterprises can quickly capture market changes and make adjustments (Singh and Hess, 2017) [11], interact with customers in real time to better respond to customer needs (Hansen and Sia, 2015) [12], create new value for customers (Yoo et al., 2010) [13], and thus improve business performance and market position (Dimitrov, 2016) [14]. As for the impact of digital transformation on the dynamic capabilities of enterprises, over time, the dynamic capabilities of enterprises to obtain information by relying on digital technology continuously improve, and the capabilities to integrate enterprise resources using information technology have been continuously optimized (Karimi and Walter, 2015; George and Schillebeeckx, 2022) [15,16]. With regard to the impact of digital transformation on enterprise stock liquidity, the digital transformation of enterprises enhances the level of enterprise stock liquidity by improving positive market expectations, promoting enterprise innovation performance, and enhancing enterprise value and financial stability (Wu Fei et al., 2021) [17]. With regard to the impact of digital transformation on earnings management, digital transformation restrains real activity earnings management by improving the company’s resource operation efficiency and information transparency, constraining managers’ self-interest motives (Luo Jinhui and Wu Yilong, 2021) [18]. With regard to the impact of digital transformation on real financialization, the improvement of digital transformation has promoted enterprise R&D investment, optimized internal control, and thus effectively curbed excessive financialization (Xu Chaohui and Wang Mansi, 2022a) [19].

Digital transformation provides advantages for enterprise value creation, but it also has destructive effects (Westerman, 2016) [20]. The digitalization process has promoted the development of the business model, but it will also increase the management expenses and labor costs. The hidden costs of the digitalization transformation of enterprises are high (Ekata, 2012) [21], meaning that digitalization does not significantly improve enterprise performance. Relying on digital technology to make decisions without paying attention to emotional factors, the effectiveness of decision suggestions based on digital derivation is questionable (Logg et al., 2019) [22]. Digital transformation has improved market rent sharing, resulting in an increase in the remuneration of senior executives and ordinary employees, but the increase in senior executives’ remuneration is even greater, which further exacerbates the internal income inequality of the enterprise (Xu Chaohui and Wang Mansi, 2022b) [23].

### 2.2. Influencing Factors of Environmental Governance

As for the definition of concepts and indicator measurements related to corporate environmental governance, the existing literature has not yet formed a unified standard, but mostly uses the concept of environmental performance. For the influencing factors of corporate environmental governance, existing research mainly focuses on external and internal motivations.

In terms of external motivations, these mainly include government (Lei et al., 2022), media (Zhang Yuming et al., 2021), and non-profit organizations (Hartmann and Uhlenbruck, 2015) [24,25,26]. The Government’s environmental law enforcement has a deterrent effect, which can reduce enterprise pollution emissions and promote enterprise compliance (Earnhart et al., 2004; Shimshack and Ward, 2008) [27,28], but it cannot urge enterprises to invest in pollution reduction technologies (Prechel and Zheng, 2012) [29]. Research by Xu Liping et al. (2018) shows that media attention can help to encourage a forward-looking environmental tone, thereby improving corporate environmental performance [30]. Yan et al. (2021) also found that increasing the proportion of green investment funds in the financial sector can effectively improve corporate environmental performance [31].

In terms of internal motivation, this mainly includes corporate governance structure, R&D investment and corporate social responsibility. Shive and Forster (2020) argue that a negative association between emissions and mutual fund ownership and board size can be found within public firms, suggesting that increased oversights may decrease externalities [32]. Wall et al. (2012) analyzed the influencing factors of corporate environmental governance from the perspective of corporate governance, and found that ownership concentration plays a positive role in promoting corporate environmental governance [33]. Alam et al. (2019) found that R&D investment can reduce energy consumption and carbon emissions and improve environmental performance [34]. Jiang et al. (2014) also discovered that R&D intensity can reduce pollutant emissions of Chinese manufacturing enterprises [35]. Kraus et al. (2020) showed that corporate social responsibility is positively related to environmental strategy and green innovation, thereby improving environmental performance [36]. Giovanni et al. (2012) revealed that the implementation of green production practices by enterprises will improve environmental performance, which is recognized by stakeholders [37].

The existing literature provides a theoretical basis for exploring the relationship between digital transformation and environmental governance, but few of studies focus on the mining enterprises with the most prominent environmental issues. Although the Government’s use of administrative power can significantly improve the enterprise’s environmental governance capability in the short term, it cannot stimulate the internal power of enterprise environmental governance. Corporate governance plays a positive role in environmental governance, but its effectiveness is low. The digital transformation of enterprises reshapes business processes, and the organizational structure and business model of enterprises, and deeply affects the production, operation and management activities of enterprises. It can not only effectively stimulate the power of enterprise environmental governance, but also plays a more superior role in promoting enterprise environmental governance.

## 3. Theoretical Analysis and Research Hypothesis

The digital transformation of enterprises refers to the process of achieving business improvement, efficiency enhancement and value creation model reshaping based on the broad fusion application of the new generation underlying digital technologies, including artificial intelligence, blockchain, cloud computing, and big data (Vial, 2019; Fischer et al.,2020) [38,39]. Corporate digitization is essentially manifested as: empower original working scenarios with information-based and coding-based features by digital technologies, and achieve real-time data acquisition, data analysis, and visual data by building the data middle-end (Luo Jinhui and Wu Yilong, 2021) [18]. Digital transformation enables enterprises to encode data and output them into standardized and structured information, thus effectively easing the information asymmetry at both internal and external levels of enterprises (Liu et al., 2011) [40]. This is convenient for strengthening media supervision and optimizing the comparability of accounting information.

Under the background of ecological civilization construction, the environmental protection of enterprises is a major focus of news media coverage and should be carefully considered by banks, investors, governments and other stakeholders. The essence of media is a kind of “information transmission” intermediary, and its function of information collection, processing and dissemination is conducive to enhancing the amount of information available to investors (Fang and Peress, 2009) [41]. With the development of information technology and the high popularity of the Internet, the role of media in information intermediation in the capital market is becoming increasingly critical (Strycharz et al., 2018) [42]. Corporate reputation is the most valuable intangible asset that can competitively benefit an enterprise (Roberts and Dowling, 2002) [43]. In order to win the favor of banks and investors and maintain their positive image, enterprises will try their best to avoid linking their brand image with environmental pollution, strengthen environmental protection investment, and realize the value of reputational assets. According to public value theory, exploring and responding to citizens’ collective preferences is the core of public decision-making processes (O’Flynn, 2007) [44]. Citizens’ recognition and satisfaction are important parts of government performance (Bao Guoxian et al., 2018) [45]. When the public learns that enterprises have environmental problems through the news media, local governments will actively take measures to promote enterprises to improve environmental performance and respond to the public’s environmental preferences and demands. Under the pressure of legitimacy, enterprises will adopt more active environmental governance behaviors to improve environmental performance (Bansal and Roth, 2000) [46]. Therefore, the digital transformation of enterprises strengthens the information environment supervised by the media, facilitates the media to comprehensively and efficiently understand and report on enterprises’ environmental issues. Moreover, banks, investors and governments can also quickly learn about an enterprise’s environmental performance through the media. In order to present a good public image regarding environmental protection work, enterprises have the motivation and pressure to increase investment in environmental protection and improve the level of environmental governance.

Accounting information comparability is equivalent to reflecting individual economic circumstances in financial reports, including equivalent financial statements and equivalent financial measurements (Simmons, 1967) [47]. This is an important information source that is necessary for information users to make decisions (Barth et al., 2013) [48], because comparable accounting information can help reduce the costs of information acquisition and processing and improve the usefulness of decisions, thus further protecting their legitimate interests. Low requirements of accounting norms on information comparability will leave greater loopholes for enterprises to exaggerate or even falsify disclosing data (Bertomeu and Ivan, 2016) [49]. Without an information disclosure system that has strict requirements for information quantification and comparability, it is possible for the enterprise to conceal environmental information with little violation cost (Doshi, 2013) [50]. If so, the enterprise will likely conceal information in the consideration of economic rationality instead of reducing pollution or increasing the investment in environmental protection. This greatly reduces the environmental governance of the enterprises, harms the interests of shareholders, and is not conducive for the long-term development of enterprises. During the digital transformation process, the use of digital technologies achieves the informatization and digitization of business data generated during the production, operation, and management processes of the enterprises. Based on these, data can further be transformed into standard financial information in the accounting information system, which takes digital technology as a foundation (Nie XingKai et al., 2022) [51]. This changes the method and efficiency of information transmission of the enterprise, improves its information environment, and enhances its accounting information comparability. The environmental information is an important part composing the contents of corporate social responsibility report. Additionally, the digital transformation actually improves the disclosure quality of non-financial information of enterprises. The more transparent the enterprise information environment, the higher the comparability level of the accounting information (De Franco, 2011) [52]. As a matter of fact, the enhancement of the accounting information comparability not only reduces information asymmetry, but also increases the violation cost of concealing information (Kim and Kraft, 2013) [53], which prompts enterprise managers to vigorously increase investment in environmental protection in order to keep conforming to legal requirements. On this basis, the following hypothesis is proposed:

**Hypothesis** **1.***Digital transformation has a significant positive correlation with the enterprise’s environmental governance, i.e., digital transformation dramatically improves the environmental governance level of the enterprise*.

## 4. Methods and Data

### 4.1. Data Source

Since 1 July 2007, Chinese listed companies have generally implemented new accounting standards; therefore, this paper selects Chinese A-share mining listed companies from 2007 to 2020 as the initial research sample. Considering the study of mining enterprises in this paper, the specific industries involved are: non-metallic mineral extraction, non-metallic mineral products, ferrous metal mining, ferrous metal smelting and rolling processing, mining auxiliary activities, coal mining and washing, oil and gas extraction, petroleum processing, coking and nuclear fuel processing, non-ferrous metal mining, non-ferrous metal smelting and rolling processing. The raw data are processed as follows: (1) exclude listed companies with ST and *ST in the sample period; (2) exclude samples with missing values of key indicators; (3) exclude samples with gearing ratio greater than 1. A total of 3208 sample observations were finally obtained for 382 companies, with data obtained from the Database of the Institute of Public and Environmental Affairs, China Energy Statistical Yearbook, CSMAR Database, and the annual reports of listed companies issued by SSE and SZSE. To control for the effect of outliers, continuous variables were winsorized at the 1% quantile.

### 4.2. Variables Selection

Explanatory variables. Enterprise digital transformation (*Digit*). Following Wu et al. (2021) [17], we took the digital technology word frequency in companies’ annual reports as a proxy indicator of the degree of digital transformation. Corporate digital transformation is an important strategy for enterprise development, and it tends to be reflected in the annual reports. We first collected a characteristic word spectrum. Digital transformation can be divided into two levels: “underlying technology application” and “technology practice application”. Underlying technology application refers to the digital-technology-driven transformation and digitalization of original technology and production systems, which rely on the layout and development of key core technologies. Among them, technologies such as AI, blockchain, cloud computing, and big data constitute the core, mainly focusing on the digital transformation of production, operation, management, and support technologies. In this regard, the characteristic word spectrum in this regard included “investment decision aid system”, “intelligent data analysis”, “data mining”, “data visualization”, “Internet of Things”, “information physical system”, and “distributed computing”. Technology practice application, meanwhile, refers to the digital transformation of enterprises focusing on the integration and innovation of digital technologies and complex business scenarios. The feature word spectrum included “intelligent marketing”, “digital marketing”, “e-commerce”, “intelligent customer service”, “B2B”, “B2C”, “C2B”, and “O2O”. The data pool was formed by extracting the text of annual reports based on Python. The summed word spectrum was formed by searching, matching, and counting the word spectrum based on the collected feature word spectrum. Finally, the natural logarithm was taken by adding 1 to the summed word spectrum and expressed by *Digit*. The larger the *Digit* value, the higher the degree of digital transformation of the enterprise.

Explained variables. Enterprise environmental governance (*EG*). There is no unified standard for the measurement of corporate environmental governance in the academic community, and the existing literature mainly uses two approaches to measure corporate environmental governance performance: the subjective-based environmental composite score and corporate environmental expenditure. Patten (2005) points out that the environmental capital expenditure of enterprises is a relatively accurate objective environmental performance indicator [54]. Therefore, this paper draws on the literature, such as Wang, Yun et al. (2017) to use corporate environmental capital expenditure as a proxy variable for corporate environmental governance [55]. The disclosure of environmental protection investment data of listed companies mainly appears in the environmental and sustainable development section of CSR reports, including investments in environmental protection technology improvement projects, pollution treatment investment, environmental protection facility renovation, operation and management, sewage fee payment, and cleaner production. The total amount of each enterprise’s environmental protection investment was screened by hand. For environmental protection investment in millions of CNY, the natural logarithm is taken and expressed as *EG*. The larger the value of this indicator, the better the environmental governance performance.

Controlled variables. In addition to digital transformation, there are many factors that influence corporate environmental governance, and the effects of firm characteristics and corporate governance are controlled with reference to existing studies in the literature. Among them, company characteristics are mainly profitability (measured by return on net assets; denoted by *ROA*), financial leverage (measured by gearing ratio; denoted by *Lev*), company growth (measured by growth rate of company assets; denoted by *Agr*), company cash flow (ratio of net cash flow from company operating activities to total assets; denoted by *CF*), nature of property rights (state-owned enterprises take the value of 1), and private enterprises (expressed by *SOE*); corporate governance, mainly the governance of major shareholders (measured by the shareholding ratio of the largest shareholder; expressed by *Ls*); the governance of independent directors (measured by the number of independent directors in the total number of directors; expressed by *Id*); and dual positions (the value of 1 when the general manager is also the chairman; otherwise, the value of 0, expressed by *Pt*). The variables and descriptions used in this paper are shown in Table 1.

### 4.3. Model Specification

To test the research hypothesis, a linear regression model was constructed as follows:(1)$$EG_{i,t}=c_{0}+\alpha_{1}Digit_{i,t}+\alpha_{i}Control_{i,t}+\sum Industry+\sum Year+\varepsilon_{i,t}$$

The explained variable in regression Model (1) is enterprise environmental governance (*EG*), the core explanatory variable is digital transformation (*Digit*), *Control* is an above-mentioned controlled variable, and ε is a random error. When the regression coefficient $\alpha_{1}$ in Model (1) is a significant positive, this indicates that digital transformation improves corporate environmental governance.

## 5. Empirical Results and Analysis

### 5.1. Descriptive Statistics

Descriptive statistics for the main variables are presented in Table 2. Table 2 shows the descriptive statistics of the main variables. Digital transformation (*Digit*) has a mean value of 0.3374 and a median value of 0, indicating that more than half of the mining companies have not implemented a digital transformation strategy. Environmental governance (*EG*) has a mean value of 15.6731 and a median value of 17.6972, which indicates that mining companies do not show much variation in their investment in environmental protection. The mean value of profitability (*ROA*) is 0.0364, and the median value is 0.0306, which indicates that the profitability of mining companies is generally low. The financial leverage (*Lev*) has a mean of 0.4838, median of 0.4945 and standard deviation of 0.2032, indicating that the financial leverage of mining companies is generally high. Growth (*Agr*) has a mean value of 0.1731 and a median value of 0.0806, indicating that the growth of mining companies is generally low. The mean value of cash flow from operating activities (*CF*) is 0.0542 and the median value is 0.0502, which indicates that mining companies have low cash flow from operating activities. The nature of ownership (*SOE*) has a mean value of 0.5527 and a median value of 1, indicating that the majority of mining companies are state-owned enterprises. The mean value of majority shareholder governance (*Ls*) is 0.3913, the median value is 0.3812 and the standard deviation is 0.1689, indicating that the phenomenon of “one share dominance” is more common in mining companies. The mean value of independent director governance (*Id*) is 0.3693 and the median value is 0.3333, indicating that the number of independent directors in mining companies generally meets the statutory requirements. The mean value of general manager and president served by one person is 0.1761 and the median is 0, meaning that less than 50% of the enterprises have the situation of general manager and president served by one person.

In Table 3, the correlation coefficient between digital transformation (*Digit*) and environmental governance (*EG*) is 0.043, which is significant at the 5% level, indicating that digital transformation significantly enhances corporate environmental governance, validating the accuracy of the previous hypothesis to some extent. The correlation coefficients between the variables are generally less than 0.8, which indicates that there is less possibility of cointegration when linear regressions are conducted in the latter.

### 5.2. Empirical Analysis

Table 4 shows the benchmark regression results for the linear regression of Model (1). The regression coefficient of digital transformation (*Digit*) was 0.4580, and the *t*-value was 2.57 when no control variables were considered, which was significant at the 1% level. Meanwhile, the regression coefficient of digital transformation (*Digit*) was 0.3574, and the *t*-value was 2.53 when control variables were considered, which was significant at the 5% level. This indicates that digital transformation is significantly and positively correlated with the environmental governance of mining companies, and the higher the degree of digital transformation, the higher the level of environmental governance of mining companies. The previous hypothesis was verified.

Among the control variables, the regression coefficient of profitability (*ROA*) is 51.7556, which is significant at the level of 1%, indicating that mining enterprises with strong profitability will increase their investment in environmental protection. The regression coefficient of the property right nature (*SOE*) is 0.9602, which is significant at the 1% level, indicating that compared with non-state-owned mining enterprises, state-owned mining enterprises are more willing to increase environmental protection investments. The regression coefficient of major shareholders’ governance (*Ls*) is 1.3877, which is significant at the 5% level, indicating that the higher the shareholding ratio of major shareholders, the more environmental protection investment of mining enterprises.

### 5.3. Robustness Analysis

#### 5.3.1. Panel Data Fixed-Effect Model

Considering the possible endogeneity problem, the panel data fixed effects model test was used, and Table 5 shows the regression results of the panel data fixed effects model. In Table 5, the regression coefficient of digital transformation (*Digit*) is 0.3574, and the *t*-value is 2.53, which is significant at the 5% level during the fixed effects model test. Robust and Cluster robust standard error tests were also conducted. For the impact of digital transformation on environmental governance, Table 5 presents the results of Robust heteroskedasticity robust standard error and Cluster clustering robust standard error tests using digital transformation (*Digit*), where the regression coefficients are significantly positive. These regression results show consistency with previous tests.

#### 5.3.2. Variable Substitution

Table 6 shows the regression results for variable substitution. As shown in Table 6, the dummy variable (*Digita_dummy*) was used to denote the digital transformation of mining enterprises: the presence of digital transformation of mining enterprises is denoted by 1, and the absence of digital transformation of mining enterprises is denoted by 0. Then, a linear regression of Model (1) was conducted. The regression coefficient of digital transformation (*Digita_dummy*) is 0.7528, and the *t*-value is 3.46, which is significant at the 1% level significant at the 1% level, indicating that mining companies with digital transformation make a significant contribution to environmental governance compared to mining companies without digital transformation.

The digital transformation indicators of enterprises are divided into two major levels: underlying technology (*Digit_basic*) and practical application (*Digit_used*). In particular, *Digit_basic* is calculated by counting the frequency of words related to artificial intelligence technology, big data technology, cloud computing technology, and blockchain technology in the annual reports of mining companies and forming a summed word frequency, plus one by natural logarithm. The practical application level (*Digit_used*) is expressed by counting the frequency of words in the use of digital technologies in the annual reports of listed companies, plus one to take the natural logarithm. *Digit_basic* and *Digit_used* are regressed separately for environmental governance. In the path, “*Digit_basic*→*EG*”, the regression coefficient of *Digit_basic* is 0.2244 and the *t*-value is 0.98, the regression coefficient of *Digit_basic* is 0.2244 and the *t*-value is 0.98, indicating that the development of underlying technologies can contribute to environmental governance but is not significant. In the path “*Digit_used*→*EG*”, the regression coefficient of *Digit_used* is 0.4008 with a *t*-value of 2.50, which is significant at the 5% level, indicating that the practical application of digital technology in physical enterprises can significantly contribute to environmental governance. It can be seen that the digital transformation of physical enterprises can effectively contribute to environmental governance mainly by relying on the practical application of digital technologies.

#### 5.3.3. Instrumental Variables

Table 7 shows the endogeneity test of instrumental variables. The instrumental variables approach is used to further reduce endogeneity interference to enhance the robustness of the core study findings. In terms of the choice of instrumental variables, drawing on Chen-Yu Zhao et al. (2021) [56], the volume of telecommunication services (*Telecom*) and Internet broadband access ports (*Net*) in the provinces where mining companies are listed were chosen as the instrumental variables for the endogeneity test.

Firstly, the endogeneity of digital transformation variables was determined. In the first step, digital transformation (*Digit*) was regressed on all exogenous variables (*Telecom*, *Net*, *ROA*, *Agr*, *Lev*, *CF*, *SOE*, *Ls*, *Id*, *Pt*) to obtain the residual E. In the second step, the residual E is added as an explanatory variable in the original Model (1) to obtain the following model.
(2)$$EG_{i,t}=\alpha_{0}+\alpha_{1}Digit_{i,t}+\alpha_{i}Control_{i,t}+\rho E+\mu$$

Test whether the regression coefficient ρ of *E* is 0 or not. According to the test result, the regression coefficient ρ is −2.0691, indicating that it should reject the original hypothesis and consider the existence of endogeneity problem. Due to the existence of the endogeneity problem in the original model, the instrumental variable method was set to be used in this process.

Secondly, the correlation between the selected instrumental variables and the endogenous explanatory variables was examined (to determine the reliability of the instrumental variables). Endogenous variables (*Digit*) were regressed on all exogenous and instrumental variables, and the model was constructed as follows.
(3)$$Digit_{i,t}=\alpha_{0}+\alpha_{i}Control_{i,t}+\alpha_{8}Telecom+\alpha_{9}Net+\varepsilon$$

The test results found that the regression coefficient for *Telecom* was 0.0834 and the T-value was 5.00, which was significant at the 1% level. The regression coefficient for *Net* was 0.0955 and the T-value was 5.78, which was significant at the 1% level.

When estimating the constrained F-test to test whether the coefficients of the two instrumental variables are simultaneously zero, it was found that the regression coefficients of both instrumental variables are not zero and F (2, 3197) = 115.96, Prob > F = 0.0000. This indicates that the volume of telecommunication services (Telecom), Internet broadband access ports (*Net*), as instrumental variables and endogenous variables (*Digit*), are correlated and can explain part of the information of digital transformation. After finding the instrumental variables, a two-stage least squares 2sls estimation was carried out.

Finally, the method of the two-stage least squares 2sls was used to deal with the endogeneity problem. The estimator of *Digit* was used as a substitution variable to perform a regression to all exogenous variables. Based on this, Model (4) was constructed as follows.
(4)$$EG_{i,t}=\alpha_{0}+\alpha_{1}(Digit_{i,t}=Telecom\;Net)+\alpha_{i}Control_{i,t}+\mu$$

In the second-stage model regression results, as shown in Table 7, the digital transformation of enterprises still produces a significant contribution to environmental governance, and the regression coefficients pass the 1% significance level test, which indicates that the core research findings of this paper are still confirmed.

#### 5.3.4. First Difference Test

The first difference was conducted for Model (1) to relieve the endogenous problems caused by the reverse causality between digital transformation and enterprise environmental governance.

Based on the panel data regression, the impacts of digital transformation on environmental governance were tested. Table 8 shows the test results of the first difference model. When controlled variables are not considered, after the first difference was conducted for Model (1), the regression coefficient of Δ*Digit* is 0.8029 and significant at a 1% level; when controlled variables are considered, after the first difference was conducted for Model (1), the regression coefficient of Δ*Digit* is 0.7305 and significant at a 1% level. It can be seen that digital transformation can significantly inhibit enterprise environmental governance.

### 5.4. Inspection of Channel Mechanism Expansion

The above-mentioned hypothesis is validated in the benchmark regression, but the internal acting mechanism between digital transformation and environmental governance is yet to be studied.

#### 5.4.1. Channel Mechanism of Media Supervision

The communicative aspects of the media can effectively supervise the operating behaviors of enterprises by channels of spreading information of other intermediary agencies, and generating new information by individual mining (Miller, 2006) [57]. Under media communications, the environmental protection violations of mining enterprises could induce the disturbance of public opinion, which further harms the images of the enterprise and its executives. Therefore, in order to protect their reputation, enterprises would rather invest in environmental protection to reduce a public opinion crisis caused by environmental protection problems. Digital transformation breaks the information barrier, reduces the information asymmetry between enterprises and external stakeholders, and facilitates the media to collect and mine environmental protection information of enterprises. Additionally, the digital transformation of enterprises is enhancing the supervision capability of the media, further improving the level of enterprise environmental governance under the reputation mechanism.

Therefore, according to the principle of intermediary effect test, on the basis of Model (1), the pressure channels of external media supervision are tested further. The mediation effect model is set as follows:(5)$$Media_{i,t}=c_{0}+\alpha_{1}Digit_{i,t}+\alpha_{i}Control_{i,t}+\sum Industry+\sum Year+\varepsilon_{i,t}$$
(6)$$EG_{i,t}=c_{0}+\alpha_{1}Digit_{i,t}+\alpha_{2}Media_{i,t}+\alpha_{i}Control_{i,t}+\sum Industry+\sum Year+\varepsilon_{i,t}$$

Among them, media supervision is an intermediary variable.

Therefore, a further inspection on the pressure channel of external media supervision was performed according to the principle of the mediating effect test. As for the measurement of the media supervision, by referring to the method of Kim et al. (2019) [58], the quantitative statistical data of newspapers and network news disclosed in the news and public opinion database of listed companies from 2007 to 2020 of China Research Data Service Platform (CNRDS) are used in this paper, from which, a total of 3165 sample observation values are obtained. The number of reports in newspapers and online media is calculated with the addition of 1 to obtain the natural logarithm, and thus the media supervision variable is obtained (*Media*). The larger the media value, the greater the pressure of external media supervision on the mining enterprise. Table 9 shows the regression results of the media supervision channels. As shown in this table, the path “*Digit→EG*” was tested first, for which the regression coefficient of digital transformation (*Digit*) is 0.3480 and T value is 2.55, significant at a 5% level. This means that digital transformation significantly promotes environmental governance of the mining enterprises. Next, a test of the “*Digit→Media*” path was carried out: the regression coefficient of its digital transformation (*Digit*) is 0.1061, the T value is 3.76, and the test is significant at a 1% level. This indicates that digital transformation significantly promotes media supervision. Finally, the impact of digital transformation (*Digit*) and media supervision (*Media*) on environmental governance is tested, which proves that the digital transformation (*Digit*) has a 0.3102 regression coefficient and is significant at 5% level, while the regression coefficient of media supervision is 0.3562, and significant at 1% level. According to the principle of the mediating effect test, media supervision has a significant mediating effect on both digital transformation and environmental governance; digital transformation improves media supervision, and further promotes the environmental governance of mining enterprises.

#### 5.4.2. Channel Mechanism of Accounting Information Comparability

Comparability is an important feature measuring the quality of enterprise accounting information. Higher comparability of accounting information between enterprises and other peer enterprises means more transparent enterprise information, which facilitates stakeholders to obtain more comprehensive information about enterprises, thus further strengthening the supervision towards enterprise environmental problems. Digital transformation helps solve the problem of information asymmetry and strengthen the quality of internal control in enterprises. In these processes, the comparability of accounting information can be enhanced, which makes it possible to fully expose the environmental problems of mining enterprises to stakeholders and promote enterprises to improve their environmental governance.

Therefore, based on the principle of the intermediary effect test, the comparability channel of accounting information is further tested.

On the basis of Model (1), the comparability of internal accounting information is further tested. The mediation effect model is set as follows:(7)$$Infor_{i,t}=c_{0}+\alpha_{1}Digit_{i,t}+\alpha_{i}Control_{i,t}+\sum Industry+\sum Year+\varepsilon_{i,t}$$
(8)$$EG_{i,t}=c_{0}+\alpha_{1}Digit_{i,t}+\alpha_{2}Infor_{i,t}+\alpha_{i}Control_{i,t}+\sum Industry+\sum Year+\varepsilon_{i,t}$$

Among them, the comparability of accounting information is an intermediary variable.

In view of this, the comparability channel of accounting information was further tested according to the principle of mediation effect. The method of De Franco et al. (2011) was used to measure the comparability of accounting information [52]. The target companies were matched with other peer companies of the same industry one by one in order to calculate the accounting information comparability of each pair of target and peer companies. The mean value of the comparability was taken to infer the accounting information comparability of the target company, which was recorded as *Infor*. The larger the *Infor* value, the higher the comparability of the target company’s accounting information, i.e., the quality of the accounting information is higher. The financial statistics of companies listed from 2007 to 2020 in the database of China Stock Market Accounting Research (CSMAR) are used in this study, from which a total of 1821 sample observation values are obtained. Please see Table 10 for the regression results of the channel effect of accounting information comparability. In this paper, the “*Digit→Infor*” path is tested first, which shows a 0.3448 regression coefficient of digital transformation (*Digit*) and 1.72 *t*-value, with a significant performance at a 10% level. This unveils the fact that digital transformation can significantly promote the environmental governance of mining enterprises. Then, the “*Digit→Infor*” path was tested, which has a 0.0001 regression coefficient of digital transformation (*Digit*), 0.31 *t*-value, and is significant at a 1% level, indicating that digital transformation promotes the comparability of accounting information of mining enterprises but is not significant. Finally, a relative test was carried out to determine the impact of digital transformation (*Digit*) and accounting information comparability (*Infor*) on environmental governance: for digital transformation (*Digit*), the regression coefficient is 0.3342 and is significant at 10% level, while for the accounting information comparability (*Infor*), the regression coefficient is 125.3633, and is significant at 1% level. According to the principle of mediating effect test, the accounting information comparability has no significant mediating effect between digital transformation and environmental governance, i.e., digital transformation cannot promote environmental governance through accounting information comparability.

### 5.5. Heterogeneity Analysis

#### 5.5.1. Distinguish the Nature of Property Rights

Considering the particularity of the property rights of Chinese enterprises, the overall sample was divided into state-owned enterprises and non-state-owned enterprises according to property rights, and the heterogeneity characteristics of the relationship between digital transformation and environmental governance of mining enterprises in state-owned enterprises and non-state-owned enterprises were analyzed. Table 11 shows the regression results of distinguishing the nature of property rights. In state-owned enterprises, the impact of digital transformation on environmental governance of state-owned mining enterprises was analyzed. The regression coefficient of digital transformation (*Digit*) is 0.2161, and the t value is 1.04. Moreover, the impact of digital transformation on the environmental governance of non-state-owned mining enterprises was analyzed. The regression coefficient of digital transformation (*Digit*) is 0.3622, and the t value is 1.95. It can be seen that the promotion of digital transformation on environmental governance is only significant in non-state-owned enterprises. This may be because, compared with state-owned enterprises, non-state-owned enterprises have a higher degree of marketization and a faster speed of digital transformation, which makes the effect of digital transformation on environmental governance more evident in non-state-owned enterprises.

#### 5.5.2. Differentiation of Enterprise Scale

Considering the scale effect of digital transformation of enterprises, the overall sample was divided into large-scale enterprises and small- and medium-sized enterprises according to the size of enterprises. The heterogeneity characteristics of the relationship between the digital transformation and environmental governance of mining enterprises in large-scale enterprises and small and medium-sized enterprises were analyzed. Table 12 shows the regression results of distinguishing enterprise scale. In large-scale enterprises, the impact of digital transformation on environmental governance of mining enterprises is analyzed. The regression coefficient of digital transformation (*Digit*) is 0.3660 and t value is 1.80. Among small- and medium-sized enterprises, the impact of digital transformation on the environmental governance of mining enterprises is analyzed. The regression coefficient of digital transformation (*Digit*) is 0.0609, and the t value is 0.32. It can be seen that the promotion of digital transformation on environmental governance is only significant in large-scale enterprises. The reason may be that small- and medium-sized enterprises have less production and management data, and using digital technology to mine valuable information is difficult.

#### 5.5.3. Differentiate Enterprise Life Cycle

Dickinson’s (2011) method of cash flow model was used as a reference to categorize parts of the enterprise life cycle [59]. The cash flow model reflects the operating risk, profitability, growth rate and other characteristics of different life cycle stages through the positive and negative combination of net cash flows from operating activities, investment activities and financing activities. According to the cash flow characteristics of enterprises in different life cycle stages and the fact that China’s listed mining enterprises have passed the initial stage, the life cycle of China’s listed mining enterprises is divided into a growth period, maturity period and recession period. Table 13 shows the regression results of enterprise life cycle. In the growth sample, the impact of digital transformation on the environmental governance of mining enterprises was analyzed. The regression coefficient of digital transformation (*Digit*) is 0.4555, and the t value is 2.25. In the mature sample, the impact of digital transformation on environmental governance of mining enterprises was analyzed. The regression coefficient of digital transformation (*Digit*) is 0.4273, and the t value is 2.00. In the sample of recession period, the impact of digital transformation on environmental governance of mining enterprises was analyzed. The regression coefficient of digital transformation (*Digit*) is −0.1206, and the t value is −0.26. Compared with the recession period, the promotion of digital transformation on environmental governance of mining enterprises is only significant during the growth period and recession period. This may be because the mining enterprises in the recession are difficult to operate, their profitability is weak, and they lack capital resources for digital transformation. However, mining enterprises often have long-term development prospects during the growing period and are motivated to carry out digital transformation. Mature enterprises have a stable cash flow, broad financing channels, and sufficient resources for digital transformation.

## 6. Conclusions

With the prosperous development of digital technologies in recent years, such as the artificial intelligence, blockchain, cloud computing and big data, digital transformation has turned out to be a new impetus for the green development of enterprises. This paper employs a text mining method to construct digital transformation data of enterprises from the green development perspective of mining enterprises. A-share listed mining companies on the Shenzhen and Shanghai stock exchanges between 2007 and 2020 were used as research samples to empirically test the impact of digital transformation on the environmental governance of mining enterprises and relative acting mechanisms. It was found that digital transformation significantly improves the environmental governance level of the mining enterprises. The above conclusions are still valid even after a series of endogeneity and robustness tests are conducted. The path tests prove that the digital transformation of mining enterprises improves environmental governance by strengthening media supervision, but the comparability of accounting information has no significant mediating effect between the digital transformation and environmental governance. Further considering the characteristics of enterprise heterogeneity, compared with state-owned enterprises, the promotion of digital transformation in environmental governance is only significant among state-owned enterprises. Compared with small- and medium-sized enterprises, the promotion of digital transformation on environmental governance is only significant among large-scale enterprises. Compared with enterprises in recession, the promotion of digital transformation in environmental governance is only significant in the mature-growth enterprises.

The research of this paper not only provides certain reference for policy making, but also has relative practical meaning. First of all, with the development of digital economy, the digital transformation has become an important engine that promotes the green development of enterprises. In this context, China should seize the opportunity of digital transformation, strengthen the support for the digital transformation of enterprises, make good use of the digital transformation to achieve the dividend of digital transformation, thus accelerating the green development of enterprises. Second, to promote the green development of enterprises, external governance forces such as media supervision should be applied in addition to relying on internal governance force of the enterprises. It is found from this study that; the media supervision actually plays a role of bridge connecting digital transformation and environmental governance. Therefore, the media supervision should be taken as the edge tool for correcting environmental problems, and make the enterprises dare not to induce environmental problems and let them consciously follow the environmental protection rules. Third, the disclosure of corporate environmental information should be standardized to reduce information asymmetry and unpack the "black box" of corporate environmental issues. According to the study of this paper, the comparability of accounting information has no significant mediating effect between digital transformation and environmental governance, which mean that the low quality of corporate accounting information blocks the realization of full effects of digital transformation in enterprises. Therefore, it is necessary to strengthen the standardization of information disclosure.

The limitations of this paper include the following. Firstly, this paper only studies the economic consequences of enterprise digital transformation from the perspective of green development, lacking an in-depth study on the blocking factors of enterprise digital transformation. Secondly, this paper only generally studies the relationship between the digital transformation and environmental governance of mining enterprises and relative internal acting mechanisms, but fails to perform a study on the heterogeneity of both corporate and individual manager. Thirdly, digital transformation cannot improve the environmental governance through the comparability of accounting information, and thus this problem of smoothing the path between digital transformation and environmental governance remains unsolved. In view of these limitations, further research on these topics is in the pipeline.

## Figures and Tables

**Table 1 ijerph-19-16474-t001:** The meanings of parameters, variables and functions.

Symbols	Meaning
*Digit*	Search, match and count the digital feature thesaurus collected from the annual report of listed companies to form a total thesaurus, and add 1 to the total thesaurus to get the natural logarithm
*EG*	Take natural logarithm for environmental protection investment of enterprises
*ROA*	Return on Total Assets, ratio of the company’s net profit to the average total assets
*Lev*	The asset-liability rate, the ratio of total liabilities to total assets
*Agr*	Total Assets Growth Rate, the total assets growth of the year divided by total assets at the beginning of the year
*CF*	Cash flow of the company, the ratio of net cash flow from operating activities to total assets
*SOE*	The property right, the value of state-owned enterprise is 1, and the value of non-state-owned enterprise is 0
*Ls*	Large shareholder governance, the holding percentage of the largest shareholder
*Id*	The independent director governance, the proportion of independent directors in board size
*Pt*	The duality of CEO, the value of chairman concurrently as general manager is 1, otherwise the value is 0

**Table 2 ijerph-19-16474-t002:** Descriptive statistics of the main variables.

Variables	Mean	Median	Max	Min	SD	N
*Digit*	0.3374	0	4.2627	0	0.6755	3208
*EG*	15.6731	17.6972	25.6421	0	6.4856	3208
*ROA*	0.0364	0.0306	0.6271	−0.6438	0.0750	3208
*Lev*	0.4838	0.4945	0.9934	0.0071	0.2032	3208
*Agr*	0.1731	0.0806	27.9995	−0.8282	0.6449	3208
*CF*	0.0542	0.0502	0.7347	−0.7623	0.0814	3208
*SOE*	0.5527	1	1	0	0.4972	3208
*Ls*	0.3913	0.3812	0.95	0.0238	0.1689	3208
*Id*	0.3693	0.3333	0.7143	0.1111	0.0532	3208
*Pt*	0.1761	0	1	0	0.3809	3208

**Table 3 ijerph-19-16474-t003:** Pearson correlation analysis.

**Variables**	*Digit*	*EG*	*ROA*	*Lev*	*Agr*	*CF*	*SOE*	*Ls*	*Id*	*Pt*
*Digit*	1									
*EG*	0.043 **	1								
*ROA*	−0.009	0.618 ***	1							
*Lev*	−0.036 *	−0.201 ***	−0.354 ***	1						
*Agr*	−0.032 *	0.109 ***	0.190 ***	−0.104 ***	1					
*CF*	0.011	0.256 ***	0.393 ***	−0.089 ***	−0.041 **	1				
*SOE*	−0.084 ***	0.055 ***	−0.051 ***	0.290 ***	−0.095 ***	0.082 ***	1			
*Ls*	−0.057 ***	0.129 ***	0.076 ***	0.157 ***	−0.075 ***	0.151 ***	0.385 ***	1		
*Id*	0.071 ***	−0.027	−0.046 ***	−0.026	−0.019	−0.044 ***	0.020	0.052 ***	1	
*Pt*	0.046 ***	−0.047 ***	−0.013	−0.150 ***	0.047 ***	−0.045 ***	−0.300 ***	−0.178 ***	0.011	1

Note: *, **, and *** denote the significance levels at 10%, 5%, and 1%, respectively.

**Table 4 ijerph-19-16474-t004:** Benchmark regression results.

Variables	No Control Variables Are Considered		Consider Control Variables	
	OLS	Robust	OLS	Robust
*Digit*	0.4580 *** (2.57)	0.4580 *** (4.12)	0.3574 ** (2.53)	0.3574 *** (3.78)
*ROA*			51.7556 *** (36.07)	51.7556 *** (12.66)
*Lev*			−0.8278 * (−1.65)	−0.8278 (−1.31)
*Agr*			0.1192 (0.85)	0.1192 (1.03)
*CF*			−0.8423 (−0.69)	−0.8423 (−0.50)
*SOE*			0.9602 *** (4.58)	0.9602 *** (4.58)
*Ls*			1.3877 ** (2.22)	1.3877 *** (3.84)
*Id*			−2.2219 (−1.32)	−2.2219 (−1.26)
*Pt*			−0.2125 (−0.87)	−0.2125 (−0.82)
Constant	18.1799 *** (31.63)	18.1799 *** (28.18)	13.5248 *** (15.31)	13.5248 *** (11.30)
*R* ^2^	0.0802	0.0802	0.4250	0.4250
*N*	3208	3208	3208	3208

Note: *, **, and *** denote the significance levels at 10%, 5%, and 1%, respectively.

**Table 5 ijerph-19-16474-t005:** Regression Result of Panel Data Fixed Effect Model.

Variables	Fixed-Effect Model	Robust	Cluster
*Digit*	0.3574 ** (2.53)	0.3574 *** (3.78)	0.3574 * (1.80)
*ROA*	51.7556 *** (36.07)	51.7556 *** (12.66)	51.7556 *** (14.09)
*Lev*	−0.8278 * (−1.65)	−0.8278 * (−1.31)	−0.8278 (−1.00)
*Agr*	0.1192 (0.85)	0.1192 (1.03)	0.1192 (1.19)
*CF*	−0.8423 (−0.69)	−0.8423 (−0.50)	−0.8423 (−0.41)
*SOE*	0.9602 *** (4.58)	0.9602 *** (4.58)	0.9602 *** (3.29)
*Ls*	1.3877 ** (2.22)	1.3877 (1.52)	1.3877 (1.62)
*Id*	−2.2219 (−1.32)	−2.2219 (−1.26)	−2.2219 (−1.11)
*Pt*	−0.2125 (−0.87)	−0.2125 (−0.82)	−0.2125 (−0.83)
Constant	13.5248 *** (15.31)	0.9602 *** (3.84)	13.5248 *** (12.31)
*R* ^2^	0.4081	0.4250	0.4250
*N*	3208	3208	3208

Note: *, **, and *** denote the significance levels at 10%, 5%, and 1%, respectively.

**Table 6 ijerph-19-16474-t006:** Regression results of variable substitution.

Variables	*Digit_dummy*→*EG*	*Digit_basic*→*EG*	*Digit_used*→*EG*
*Digit_dummy*	0.7528 *** (3.46)		
*Digit_basic*		0.2244 (0.98)	
*Digit_used*			0.4008 ** (2.50)
*ROA*	52.0477 *** (36.43)	51.8457 *** (36.11)	51.6698 *** (35.99)
*Lev*	0.1229 (0.87)	−0.8210 (−1.63)	−0.8428 * (−1.68)
*Agr*	−0.3913 (−0.80)	0.1162 (0.83)	0.1191 (0.85)
*CF*	−0.3228 (−0.27)	−0.8351 (−0.68)	−0.8783 (−0.72)
*SOE*	0.9794 *** (4.75)	0.9484 *** (4.52)	0.9569 *** (4.56)
*Ls*	2.5888 *** (4.47)	1.3833 ** (2.21)	1.3914 ** (2.23)
*Id*	−1.4714 (−0.88)	−2.0810 (−1.23)	−2.2393 (−1.33)
*Pt*	−0.1801 (−0.74)	−0.2022 (−0.83)	−0.2044 (−0.84)
Constant	12.7414 *** (17.15)	13.4595 *** (15.23)	13.5648 *** (15.35)
*R* ^2^	0.3962	0.4240	0.4250
*N*	3208	3208	3208

Note: *, **, and *** denote the significance levels at 10%, 5%, and 1%, respectively.

**Table 7 ijerph-19-16474-t007:** Endogeneity test of instrumental variables.

Variables	Model (2)	Model (3)	Model (4)
*Telecom*		0.0834 *** (5.00)	
*Net*		0.0955 *** (5.78)	
*Digit*	2.4903 *** (4.93)		
*Digit =* (*Telecom Net*)			2.4903 *** (4.79)
*E*	−2.0691 (−3.90)		
Control variables	Yes	Yes	Yes
Constant	12.5290 *** (16.23)	−1.2142 *** (−9.66)	12.5290 *** (15.82)
*R* ^2^	0.3985	0.0798	0.3559
*N*	3208	3208	3208

Note: *** denotes the significance level at 1%.

**Table 8 ijerph-19-16474-t008:** Regression Results of First Difference Model.

Variables	Non-Control Variables	Control Variables
Δ*Digit*	0.8029 *** (4.14)	0.7305 *** (4.17)
*ROA*		52.3883 *** (19.83)
*Lev*		0.5196 (0.57)
*Agr*		0.5058 (1.31)
*CF*		1.0692 (0.49)
*SOE*		1.0199 *** (2.71)
*Ls*		1.4327 (1.29)
*Id*		−3.8202 (−1.26)
*Pt*		0.5485 (1.27)
Constant	0.2986 (0.32)	−4.8852 *** (−3.01)
*R* ^2^	0.0158	0.2043
*N*	3208	3208

Note: *** denotes the significance level at 1%.

**Table 9 ijerph-19-16474-t009:** Regression results of the media supervision channel effect.

Variables	*EG*	*Media*	*EG*
*Digit*	0.3480 ** (2.55)	0.1061 *** (3.76)	0.3102 ** (2.27)
*Media*			0.3562 *** (4.13)
*ROA*	53.7213 *** (38.66)	2.1637 *** (7.54)	52.9506 *** (37.86)
*Lev*	−0.3774 (−0.77)	0.4228 *** (4.20)	−0.5280 (−1.08)
*Agr*	0.0754 (0.56)	0.0758 *** (2.71)	0.0485 (0.36)
*CF*	−2.2173 (−1.36)	0.7770 ** (2.31)	−2.4941 (−1.53)
*SOE*	−0.1147 (−0.10)	0.5387 ** (2.20)	−0.3065 (−0.26)
*Ls*	2.0955 *** (3.44)	0.5819 *** (4.63)	1.8882 *** (3.10)
*Id*	−0.0879 (−0.37)	0.0396 (0.81)	−0.1021 (−0.43)
*Pt*	0.8741 *** (4.28)	0.1571 *** (3.73)	0.8181 *** (4.01)
Constant	13.4775 *** (15.74)	2.2777 *** (12.88)	12.6661 *** (14.45)
*R* ^2^	0.4585	0.2745	0.4614
*N*	3165	3165	3165

Note: ** and *** denote the significance levels at 5% and 1%, respectively.

**Table 10 ijerph-19-16474-t010:** Regression results of effects of accounting information comparability channel.

Variables	*EG*	*Infor*	*EG*
*Digit*	0.3448 * (1.72)	0.0001 (0.31)	0.3342 * (1.70)
*Infor*			125.3633 *** (7.44)
*ROA*	77.5673 *** (30.87)	0.0166 *** (4.76)	75.4920 *** (30.30)
*Lev*	1.1953 * (1.74)	−0.0112 *** (−11.82)	2.6012 *** (3.71)
*Agr*	−0.1577 (−0.23)	0.0032 *** (3.44)	−0.5607 (−0.84)
*CF*	−2.4705 (−1.32)	−0.0131 *** (−5.07)	−0.8225 (−0.44)
*SOE*	0.9240 *** (3.47)	0.0006 (1.59)	0.8506 *** (3.24)
*Ls*	1.8537 ** (2.19)	0.0004 (0.34)	1.8039 ** (2.16)
*Id*	−1.9514 (−0.84)	−0.0033 (−1.04)	−1.5340 (−0.67)
*Pt*	−0.3105 (−0.91)	−0.0003 (−0.69)	−0.2695 (−0.80)
Constant	11.5481 *** (9.86)	−0.0092 *** (−5.68)	12.7014 *** (10.91)
*R* ^2^	0.5222	0.3661	0.5366
*N*	1821	1821	1821

Note: *, **, and *** denote the significance levels at 10%, 5%, and 1%, respectively.

**Table 11 ijerph-19-16474-t011:** Regression results of distinguishing property rights.

Variables	State-Owned Enterprise	Non-State-Owned Enterprises
*Digit*	0.2161 (1.04)	0.3622 * (1.95)
*ROA*	62.5845 *** (27.10)	44.3202 *** (24.78)
*Lev*	0.1902 (0.26)	−1.2915 * (−1.89)
*Agr*	0.6676 * (1.70)	0.0737 (0.51)
*CF*	0.6060 (0.35)	−3.5207 ** (−2.07)
*Ls*	1.9108 ** (2.22)	−0.4621 (−0.49)
*Id*	0.0788 (0.04)	−7.0834 *** (−2.69)
*Pt*	0.2662 (0.58)	−0.3915 (−1.41)
Constant	11.1720 *** (9.89)	18.6698 *** (11.66)
*R* ^2^	0.4861	0.4026
*N*	1773	1435

Note: *, **, and *** denote the significance levels at 10%, 5%, and 1%, respectively.

**Table 12 ijerph-19-16474-t012:** Regression results of distinguishing enterprise scale.

Variables	Large Scale Enterprises	Small and Medium-Sized Enterprises
*Digit*	0.3660 * (1.80)	0.0609 (0.32)
*ROA*	58.3837 *** (19.66)	47.7581 *** (29.85)
*Lev*	−1.4682 (−1.46)	−2.0400 *** (−3.40)
*Agr*	0.5192 (1.26)	0.0297 (0.21)
*CF*	−4.2583 * (−1.77)	−2.3468 * (−1.67)
*Ls*	2.2384 ** (2.19)	−0.5725 (−0.70)
*Id*	−0.0226 (−0.01)	−3.3416 (−1.51)
*Pt*	−0.2167 (−0.45)	0.0688 (0.25)
*SOE*	0.5842 * (1.70)	0.1656 (0.61)
Constant	13.7024 *** (9.24)	14.8358 *** (12.33)
*R* ^2^	0.4419	0.4370
*N*	1448	1760

Note: *, **, and *** denote the significance levels at 10%, 5%, and 1%, respectively.

**Table 13 ijerph-19-16474-t013:** Regression results of differentiating enterprise life cycle.

Variables	Growth Period	Mature Period	Recession Period
*Digit*	0.4555 ** (2.25)	0.4273 ** (2.00)	−0.1206 (−0.26)
*ROA*	57.2231 *** (26.64)	48.6666 *** (19.08)	48.3616 *** (11.77)
*Lev*	0.6907 (0.93)	−0.9940 (−1.24)	−4.1144 *** (−2.97)
*Agr*	−0.0328 (−0.23)	0.3351 (0.37)	1.8402 (1.31)
*CF*	2.2018 (1.17)	−12.7396 *** (−5.13)	5.4274 (1.60)
*Ls*	0.5420 (0.60)	0.5794 (0.61)	3.8472 * (1.78)
*Id*	0.5126 (0.21)	−5.3595 ** (−2.08)	−3.4449 (−0.62)
*Pt*	−0.0589 (−0.18)	−0.7307 * (−1.88)	0.0509 (0.06)
*SOE*	0.8140 *** (2.75)	1.2711 *** (3.89)	0.1969 (0.27)
Constant	12.1121 *** (9.53)	16.4129 *** (12.06)	13.6774 *** (4.61)
*R* ^2^	0.4407	0.4053	0.5038
*N*	1595	1253	360

Note: *, **, and *** denote the significance levels at 10%, 5%, and 1%, respectively.

## Data Availability

Not applicable.

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
