# Peer review of "Effects of Digital Transformation on Environmental Governance of Mining Enterprises: Evidence from China"

_ijerph, 2022, doi:10.3390/ijerph192416474_

Round 1
Reviewer 1 Report
This paper studies the impact of digital transformation on environmental governance of mining enterprises, and deeply analyzes the internal mechanism of the two. This is interesting. However, the main shortcomings of the study are:
1)Chinese enterprises have special institutional backgrounds. It is suggested to distinguish the nature of enterprise property rights and observe whether the relationship between digital transformation and environmental governance is different between state-owned enterprises and non-state-owned enterprises.
2)In empirical analysis, we should not only analyze the relationship between explanatory variables and explained variables, but also analyze the relationship between control variables and explained variables.
3)In the robustness test, it is suggested that further test the impact of digital transformation on environmental governance of mining enterprises as a dummy variable.
4)There are few references in the past three years, so it is recommended to supplement the latest papers of this study.
5)There are some expressions are not native English, so it is suggested to edit language with professionals.
Author Response
Comments 1: Chinese enterprises have special institutional backgrounds. It is suggested to distinguish the nature of enterprise property rights and observe whether the relationship between digital transformation and environmental governance is different between state-owned enterprises and non-state-owned enterprises.
Responses and a summary of revisions:
Thank you for your suggestions. Heterogeneity analysis has been added in the paper, which considers the nature of enterprise property rights, enterprise life cycle and enterprise scale respectively.
Comments 2: In empirical analysis, we should not only analyze the relationship between explanatory variables and explained variables, but also analyze the relationship between control variables and explained variables.
Responses and a summary of revisions:
Thank you for your suggestions. In the benchmark regression analysis, the relationship between the control variables and the explained variable have been analyzed.
Comments 3: In the robustness test, it is suggested that further test the impact of digital transformation on environmental governance of mining enterprises as a dummy variable.
Responses and a summary of revisions:
Thank you for your suggestions. According to the requirements, whether the digital transformation of mining enterprises is set as a dummy variable, and the relationship between digital transformation and environmental governance is further analyzed.
Comments 4: There are few references in the past three years, so it is recommended to supplement the latest papers of this study.
Responses and a summary of revisions:
Thank you for your suggestions. In the introduction and literature review, references in the last three years have been added.
Comments 5: There are some expressions are not native English, so it is suggested to edit language with professionals.
Responses and a summary of revisions:
Thank you for your suggestions. The paper has been polished and modified by MDPI Press.
Reviewer 2 Report
The Abstract must report the aim of the study, the basic information on the sample (time span, countries analyzed), the empirical methodology used, the main findings, and the relevant policy implications.
Introduction and Literature Review should be split into two different sections.
The Introduction should highlight the relevance of the topic, the novelty of the results, the importance of policy implications, the sample’s choice, the methodology’s appropriateness, the data used, the contribution to the literature, and the limitations of the study.
The literature review is partial and incomplete, and some recent and relevant contributions should be cited and discussed: i.e., 10.1016/j.retrec.2020.100947; 10.1016/j.jclepro.2020.123293; 10.1007/s11356-020-08180-x; 10.1016/j.renene.2020.11.050.
Delete useless citations:
The theoretical framework should be discussed more in detail.
The estimated model must be justified in light of the literature on this specific topic.
Descriptive statistics are absent.
Diagnostic tests are absent.
Robustness checks are absent.
The results should be discussed more in detail.
Comparisons with previous studies are absent.
Conclusions are too short.
Policy implications are weak.
Further research should be indicated.
Limitations of the study are not provided.
Proofreading by a native speaker is required.
The editing does not follow the journal’s guidelines.
The originality value of the study is limited.
This is a basic econometric exercise without a clear innovative intuition.
How does the paper enrich the knowledge of the scientific community?
Author Response
Comments 1: The Abstract must report the aim of the study, the basic information on the sample (time span, countries analyzed), the empirical methodology used, the main findings, and the relevant policy implications.
Responses and a summary of revisions:
Thank you for your suggestions. According to the requirements, the aim of the study, sample time span, sample analysis country, research methods, research conclusions and policy significance have been completely supplemented in the summary (line 10-28).
Comments 2: Introduction and Literature Review should be split into two different sections.
Responses and a summary of revisions:
Thank you for your suggestions. According to the requirements, the introduction and literature review have been elaborated respectively, and the literature review has been supplemented (for literature review-line 151-238).
Comments 3: The Introduction should highlight the relevance of the topic, the novelty of the results, the importance of policy implications, the sample’s choice, the methodology’s appropriateness, the data used, the contribution to the literature, and the limitations of the study.
Responses and a summary of revisions:
Thank you for your suggestions. According to the requirements, the Introduction has been highlighted the relevance of the topic, the novelty of the results, the importance of policy implications, the sample’s choice, the methodology’s appropriateness, the data used, the contribution to the literature. The limitations of the study are described at the end of the paper.
Comments 4: The literature review is partial and incomplete, and some recent and relevant contributions should be cited and discussed: i.e., 10.1016/j.retrec.2020.100947; 10.1016/j.jclepro.2020.123293; 10.1007/s11356-020-08180-x; 10.1016/j.renene.2020.11.050. Delete useless citations.
Responses and a summary of revisions:
Thank you for your suggestions. According to the requirements, in the literature review section, a large number of literatures have been supplemented, especially those in the last three years. such as Cosimo & Marco (2020), Edmund et al. (2020), Marco & Cosimo (2020), Cosimo et al. (2020) (line 41, line 43, line 56).
Comments 5: The theoretical framework should be discussed more in detail. The results should be discussed more in detail.
Responses and a summary of revisions:
Thank you for your suggestions. A detailed study has been carried out as required, further considering such factors as the nature of enterprise property rights, size and life cycle of the enterprise, and discussing the conclusions (line 698-763).
Comments 6: The estimated model must be justified in light of the literature on this specific topic.
Responses and a summary of revisions:
Thank you for your suggestions. This paper mainly studies the impact of digital transformation on environmental governance, using a multiple linear regression model. In analyzing the mechanism of digital transformation on environmental governance, the intermediary effect model is used.
Comments 7: Descriptive statistics are absent. Diagnostic tests are absent. Robustness checks are absent.
Responses and a summary of revisions:
Thank you for your suggestions. Descriptive statistics are listed and explained in table 2. Table 3 is Pearson correlation analysis, which is a diagnostic test and provides a basis for regression analysis in the following text. Robustness analysis is supplemented in Table 4.
Comments 8: Comparisons with previous studies are absent.
Responses and a summary of revisions:
Thank you for your suggestions. In the part of literature review, the existing literature is compared and analyzed, and the research direction of this paper is proposed.
Comments 9: Conclusions are too short. Policy implications are weak. Further research should be indicated. Limitations of the study are not provided.
Responses and a summary of revisions:
Thank you for your suggestions. These issues are supplemented in the research conclusion (line 779-785, line 807-819).
Comments 10: Proofreading by a native speaker is required. The editing does not follow the journal’s guidelines.
Responses and a summary of revisions:
Thank you for your suggestions. The paper has been polished and modified by MDPI Press. The editing has been followed by the journal’s guidelines.
Comments 11: The originality value of the study is limited. How does the paper enrich the knowledge of the scientific community? This is a basic econometric exercise without a clear innovative intuition.
Responses and a summary of revisions:
Thank you for your suggestions. In the introduction and literature review, the innovation points of this paper are described (line 86-150, line 227-238).

Reviewer 3 Report
Good morning
After a point, please leave 1 space. Line 142.
The introduction needs to have reference with informs and author. China is not a reference, also, there are sentences without references.
Theoretical part need to included information that is in methodology.
You should include previously studies.
You wrote about purpose, but please make sure you have an objective.
Variables selection were not clear, too theoretical, but if you do a table they can see better. (line 219) You wrote 1. There is not 2.
Please include DOI in references.
Conclusions are about results or findings you should have discussion and also, mentioned authors from theoretical review.
Author Response
Comments 1: After a point, please leave 1 space. Line 142.
Responses and a summary of revisions:
Thank you for your suggestions. As required, the sentence end of line 142 has been left 1 space, similar issues in other parts have been revised also.
Comments 2: The introduction needs to have reference with informs and author. China is not a reference, also, there are sentences without references.
Responses and a summary of revisions:
Thank you for your suggestions. Some statements have added references as required. At the same time, the introduction mainly introduces the realistic background of China's mining enterprises' digital transformation and environmental governance, this part mainly states the facts, so there are few literatures cited (line 41, line 43, line 56, line 66).
Comments 3: Theoretical part need to included information that is in methodology.
Responses and a summary of revisions:
Thank you for your suggestions. In the mechanism inspection part, the intermediary effect model analysis method has been added (line 613, line 614, line 661, line 662).
Comments 4: You should include previously studies.
Responses and a summary of revisions:
Thank you for your suggestions. The research content of literature review has been added (line 151-238).
Comments 5: You wrote about purpose, but please make sure you have an objective.
Responses and a summary of revisions:
Thank you for your suggestions. The research objectives are described in the summary and introduction.
Comments 6: Variables selection were not clear, too theoretical, but if you do a table they can see better. (line 219) You wrote 1. There is not 2.
Responses and a summary of revisions:
Thank you for your suggestions. Variables have been classified and represented in tables as required. 1 of line 219 has been deleted (line 396).
Comments 7: Please include DOI in references.
Responses and a summary of revisions:
Thank you for your suggestions. According to the requirements of the journal 《International Journal of Environmental Research and Public Health》, DOI is not required.
Comments 8: Conclusions are about results or findings you should have discussion and also, mentioned authors from theoretical review.
Responses and a summary of revisions:
Thank you for your suggestions. The research conclusions have been supplemented and discussed (line 779-785, line 807-819).

Round 2
Reviewer 3 Report
Please include DOI in references
Author Response
Thank you for your valuable suggestions. The doi has been added to the references, but there are a few documents that can't find doi.
References
- Magazzino, C., Mele, M. On the relationship between transportation infrastructure and economic development in China. Research in Transportation Economics, 2020, 88, 100947. DOI: https://doi.org/10.1016/j.retrec.2020.100947.
- Udemba E N, Magazzino C, Bekun F V. Modeling the nexus between pollutant emission, energy consumption, foreign direct investment, and economic growth: new insights from China. Environmental science and pollution research, 2020, 27, 17831-17842. DOI: https://doi.org/10.1007/s11356-020-08180-x.
- Mele M, Magazzino C. A machine learning analysis of the relationship among iron and steel industries, air pollution, and economic growth in China. Journal of Cleaner Production, 2020, 277, 123293. DOI: https://doi.org/10.1016/j.jclepro.2020.123293.
- Magazzino C, Mele M, Schneider N. A Machine Learning approach on the relationship among solar and wind energy production, coal consumption, GDP, and CO2 emissions. Renewable Energy, 2020, 151, 829-836. DOI: https://doi.org/10.1016/j.renene.2020.11.050.
- Cho, C. H., & Patten, D. M. The role of environmental disclosures as tools of legitimacy: A research note. Accounting, organizations and society, 2007, 32, 639-647. DOI: https://doi.org/10.1016/j.aos.2006.09.009.
- Wang Mansi & Xu Chaohui. The bank claims, internal governance and corporate innovation--Evidence from 2006-2015 technology intensive A listed firms. Accounting Research, 2018, 3, 42-49.
- Hinings, B., Gegenhuber, T., & Greenwood, R. (2018). Digital innovation and transformation: An institutional perspective. Information and Organization, 2018, 28, 52-61. DOI: https://doi.org/10.1016/j.infoandorg.2018.02.004.
- Yeow, A., Soh, C., Hansen, R. Aligning with new digital strategy: A dynamic capabilities approach. The Journal of Strategic Information Systems, 2017, 27, 43-58. DOI: https://doi.org/10.1016/j.jsis.2017.09.001.
- Lucas Jr, Henry, et al. Impactful research on transformational information technology: An opportunity to inform new audiences. MIS Quarterly, 2013, 37, 371-382. DOI: https://doi.org/10.2753/MIS0742-1222300100.
- Louridas, P., Ebert, C. Machine Learning. IEEE Software, 2016, 33, 110-115. DOI: https://doi.org/10.1109/MS.2016.114.
- Singh, A., Hess, T. How chief digital officers promote the digital transformation of their companies. MIS Quarterly Executive, 2017, 16, 1-17. DOI: https://doi.org/10.4324/9780429286797-9.
- Hansen, R., Sia, S. K. Hummel's digital transformation toward omnichannel retailing: Key lessons learned. MIS Quarterly Executive, 2015, 14, 51-66.
- Yoo, Y., Henfridsson, O., Lyytinen, K. Research commentary—the new organizing logic of digital innovation: an agenda for information systems research. Information systems research, 2010, 21, 724-735. DOI: https://doi.org/10.1287/isre.1100.0322.
- Dimitrov, D. V. Medical internet of things and big data in healthcare. Healthcare informatics research, 2016, 22, 156-163. DOI: https://doi.org/10.4258/hir.2016.22.3.156.
- Karimi J, Walter Z. The role of dynamic capabilities in responding to digital disruption: A factor-based study of the newspaper industry. Journal of Management Information Systems, 2015, 32, 39-81. DOI: https://doi.org/10.1080/07421222.2015.1029380.
- George, G., Schillebeeckx, S. J. D. Digital transformation, sustainability, and purpose in the multinational enterprise. Journal of World Business, 2022, 57, 101326. DOI: https://doi.org/10.1016/j.jwb.2022.101326.
- Wu Fei,Hu Huizhi,Lin Huiyan and Ren Xiaoyi. (2021). Enterprise digital transformation and capital market performance: Empirical evidence from stock liquidity. Management World, 2021, 7, 130-144.
- Luo Jinhui, Wu Yilong. Level of digital operation and real earnings management. Journal of Management Science, 2021, 34, 3-18.
- Xu Chaohui & Wang Mansi. Research on the governance effect of digital transformation on the excessive financialization of real enterprises. Securities Market Herald, 2022, 7, 23-35.
- Westerman, G. Why digital transformation needs a heart. MIT Sloan Management Review, 2016, 58, 19-21.
- Ekata, G. E. The IT productivity paradox: Evidence from the Nigerian banking industry. The Electronic Journal of Information Systems in Developing Countries, 2012, 51, 1-25. DOI: https://doi.org/10.1002/j.1681-4835.2012.tb00361.x.
- Logg, J. M., Minson, J. A., Moore, D. A. Algorithm appreciation: People prefer algorithmic to human judgment. Organizational Behavior and Human Decision Processes, 2019, 151, 90-103. DOI: https://doi.org/10.1016/j.obhdp.2018.12.005.
- Xu Chaohui, Wang Mansi. Impact of enterprise digital transformation on employee compensation. China Soft Science, 2022, 9, 108-119.
- Lei, Z., Huang, L., Cai, Y. Can environmental tax bring strong porter effect? Evidence from Chinese Listed Companies. Environmental Science and Pollution Research, 2022, 29, 32246-32260. DOI: https://doi.org/10.21203/rs.3.rs-612715/v1.
- Zhang Yuming, Xing Chao, Zhang Yu. The impact of media coverage on green technology innovation of high-polluting enterprises. Chinese Journal of Management, 2021, 18, 557-568.
- Hartmann, J., Uhlenbruck, K. National institutional antecedents to corporate environmental performance. J World Bus, 2015, 50, 729–741. DOI: https://doi.org/10.1016/j.jwb.2015.02.001.
- Earnhart, Dietrich. Panel Data Analysis of Regulatory Factors Shaping Environmental Performance. Review of Economics and Statistics, 2004, 86, 391-401. DOI: https://doi.org/10.1162/00346530432302389.
- Shimshack, J. P., Ward, M. B. Enforcement and over-compliance. Journal of Environmental Economics and Management, 2008, 55, 90-105. DOI: https://doi.org/10.1016/j.jeem.2007.05.003.
- Prechel, H., Zheng, L. Corporate characteristics, political embeddedness and environmental pollution by large US corporations. Social Forces, 2012, 90, 947-970. DOI: https://doi.org/10.1093/sf/sor026.
- Xu Li-ping, Chen Li, Zhang Shu-xia, Liu Ning. Tone at the Top Management,Media Attention and Environmental Performance. East China Economic Management, 2018, 32, 114-123.
- Yan, S., Almandoz, J., Ferraro, F. The impact of logic (in) compatibility: Green investing, state policy, and corporate environmental performance. Administrative Science Quarterly, 2021, 66, 903-944. DOI: https://doi.org/10.1177/00018392211005756.
- Shive, S. A., Forster, M. M. Corporate governance and pollution externalities of public and private firms. The Review of Financial Studies, 2020, 33, 1296-1330. DOI: https://doi.org/10.1093/rfs/hhz079.
- Walls, J. L., Berrone, P., Phan, P. H. Corporate governance and environmental performance: Is there really a link?. Strategic management journal, 2012, 33, 885-913. DOI: https://doi.org/10.1002/smj.1952.
- Alam, M. S., Atif, M., Chien-Chi, C., et al. Does corporate R&D investment affect firm environmental performance? Evidence from G-6 countries. Energy Economics, 2019, 78, 401-411. DOI: https://doi.org/10.1016/j.eneco.2018.11.031.
- Jiang, L., Lin, C., Lin, P. The determinants of pollution levels: Firm-level evidence from Chinese manufacturing. Journal of Comparative Economics, 2014, 42, 118-142. DOI: https://doi.org/10.1016/j.jce.2013.07.007.
- Kraus, S., Rehman, S. U., García, F. J. S. Corporate social responsibility and environmental performance: The mediating role of environmental strategy and green innovation. Technological Forecasting and Social Change, 2020, 160, 120262. DOI: https://doi.org/10.1016/j.techfore.2020.120262.
- De Giovanni, P., Vinzi, V. E. Covariance versus component-based estimations of performance in green supply chain management. International Journal of Production Economics, 2012, 135, 907-916. DOI: https://doi.org/10.1016/j.ijpe.2011.11.001.
- Vial, G. Understanding digital transformation: A review and a research agenda. Journal of Strategic Information Systems, 2019, 28, 118–144. DOI: https://doi.org/10.1016/j.jsis.2019.01.003.
- Fischer, M., Imgrund, F., Janiesch, C., & Winkelmann, A. Strategy archetypes for digital transformation: Defining meta objectives using business process management. Information & Management, 2020, 57, 103262. DOI: https://doi.org/10.1016/j.im.2019.103262.
- Liu, D. Y., Chen, S. W., & Chou, T. C. Resource fit in digital transformation: Lessons learned from the CBC Bank global e‐banking project. Management Decision, 2011, 49, 1728-1742. DOI: https://doi.org/10.1108/00251741111183852.
- Fang, L., & Peress, J. Media coverage and the cross‐section of stock returns. The Journal of Finance, 2009, 64, 2023-2052. DOI: https://doi.org/10.1111/j.1540-6261.2009.01493.x.
- Strycharz, J., Strauss, N., & Trilling, D. The role of media coverage in explaining stock market fluctuations: Insights for strategic financial communication. International Journal of Strategic Communication, 2018, 12, 67-85. DOI: https://doi.org/10.1080/1553118X.2017.1378220.
- Roberts, P. W., & Dowling, G. R. Corporate reputation and sustained superior financial performance. Strategic management journal, 2002, 23, 1077-1093. DOI: https://doi.org/10.1002/smj.274.
- O'flynn, J. From new public management to public value: Paradigmatic change and managerial implications. Australian journal of public administration, 2007, 66, 353-366. DOI: https://doi.org/10.1111/j.1467-8500.2007.00545.x.
- Bao Guo-xian, Bao Hai-xu, Zhang Guo-xing. Theoretical study on China’s environmental performance governance systems. China Soft Science, 2018, 6, 181-192.
- Bansal, P., & Roth, K. Why companies go green: A model of ecological responsiveness. Academy of management journal, 2000, 43, 717-736. DOI: https://doi.org/10.2307/1556363.
- Simmons, J. K. A concept of comparability in financial reporting. The accounting review, 1967, 42, 680-692.
- Barth, J. R., Lin, C., Ma, Y., Seade, J., & Song, F. M. Do bank regulation, supervision and monitoring enhance or impede bank efficiency? Journal of Banking & Finance, 2013, 37, 2879-2892. DOI: https://doi.org/10.1016/j.jbankfin.2013.04.030.
- Bertomeu, J., & Marinovic, I. A theory of hard and soft information. The Accounting Review, 2016, 91, 1-20. DOI: https://doi.org/10.2308/accr-51102.
- Doshi, A. R., Dowell, G. W., & Toffel, M. W. How firms respond to mandatory information disclosure. Strategic Management Journal, 2013, 34, 1209-1231. DOI: https://doi.org/10.1002/smj.2055.
- Nie Xingkai et al. Does enterprise digital transformation affect accounting comparability? Accounting Research, 2022, 5, 17-39.
- De Franco, G., Kothari, S. P., & Verdi, R. S. The benefits of financial statement comparability. Journal of Accounting research, 2011, 49, 895-931. DOI: https://doi.org/10.1111/j.1475-679x.2011.00415.x.
- Kim, S., Kraft, P., & Ryan, S. G. Financial statement comparability and credit risk. Review of Accounting Studies, 2013, 18, 783-823. DOI: https://doi.org/10.2139/SSRN.2094637.
- Patten, D. M. The accuracy of financial report projections of future environmental capital expenditures: a research note. Accounting, Organizations and Society, 2005, 30, 457-468. DOI: https://doi.org/10.1016/j.aos.2004.06.001.
- Wang Yun, Li Yanxi, Ma Zhuang, Song Jinbo. Media coverage, environmental regulation and corporate environment behavior. Nankai Business Review, 2017, 20, 83-94.
- Zhao Chenyu, Wang Wenchun, Li Xuesong. How does digital transformation affect the total factor productivity of enterprises? Finance & Trade Economics, 2021, 42, 114-129.
- Miller, G. S. The press as a watchdog for accounting fraud. Journal of Accounting Research, 2006, 44, 1001-1033. DOI: https://doi.org/10.1111/j.1475-679X.2006.00224.x.
- Kim, A., Moravec, P. L., & Dennis, A. R. Combating fake news on social media with source ratings: The effects of user and expert reputation ratings. Journal of Management Information Systems, 2019, 36, 931-968. DOI: https://doi.org/10.1080/07421222.2019.1628921.
- Dickinson, V. Cash flow patterns as a proxy for firm life cycle. The accounting review, 2011, 86, 1969-1994. DOI: https://doi.org/10.2308/accr-10130.